# Efficient Monte Carlo Tree Search via On-the-Fly State-Conditioned Action Abstraction

**Yunhyeok Kwak**[*1]     **Inwoo Hwang**[*1]     **Dooyoung Kim**[1]     **Sanghack Lee**[†1,2]     **Byoung-Tak Zhang**[†1]

[1] AI Institute, Seoul National University
[2] Graduate School of Data Science, Seoul National University

## Abstract

Monte Carlo Tree Search (MCTS) has showcased its efficacy across a broad spectrum of decision-making problems. However, its performance often degrades under vast combinatorial action space, especially where an action is composed of multiple sub-actions. In this work, we propose an action abstraction based on the compositional structure between a state and sub-actions for improving the efficiency of MCTS under a factored action space. Our method learns a latent dynamics model with an auxiliary network that captures sub-actions relevant to the transition on the current state, which we call state-conditioned action abstraction. Notably, it infers such compositional relationships from high-dimensional observations without the known environment model. During the tree traversal, our method constructs the state-conditioned action abstraction for each node *on-the-fly*, reducing the search space by discarding the exploration of redundant sub-actions. Experimental results demonstrate the superior sample efficiency of our method compared to vanilla MuZero [Schrittwieser et al., 2020], which suffers from expansive action space.

## 1 INTRODUCTION

Monte Carlo Tree Search (MCTS) gained prominence as a decision-time planning algorithm, showcasing its capability to solve complex sequential decision-making problems [Silver et al., 2016, 2017]. The core principle involves building a search tree and performing randomized simulations to assess actions, thereby guiding the selection process towards more promising choices over time. By incorporating the additional latent dynamics model into the tree search, it

---
[*]Equal contribution.
[†]Corresponding authors.

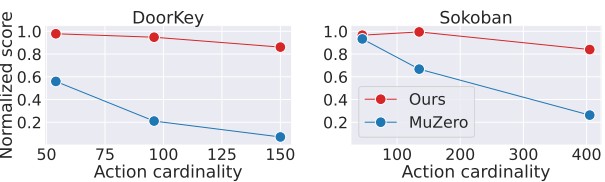

Figure 1: Normalized score of MuZero [Schrittwieser et al., 2020] and our method in environments with a factored action space. In contrast to MuZero which suffers from the increasing number of available actions, our method with state-conditioned action abstraction remains effective.

achieves remarkable performances even from pixels without the known environment model [Schrittwieser et al., 2020].

However, it often leads to sub-optimal decision-making when confronted with vast combinatorial action space. This is because the branching factor of MCTS increases as the number of available actions expands, making it challenging to efficiently explore and exploit during tree search [Couëtoux et al., 2011, Pinto and Fern, 2017, Sokota et al., 2021, Xu et al., 2022, Veeriah et al., 2022, Hoerger et al., 2023].

This issue becomes more pronounced in the environments where an action is composed of multiple sub-actions since its cardinality grows *exponentially* with respect to the number of sub-actions, as illustrated in Fig. 1. Unfortunately, such a factorized action structure is prevalent in many real-world applications. For instance, in the context of recommender systems, an action consists of multiple recommendations on a single page. In healthcare, configurations of various medications and treatments constitute an action. Many classical domains also involve factored action space, e.g., arcade games where the players manipulate multiple controllers such as joysticks and buttons simultaneously.

Existing approaches to extend MCTS to environments with factored action space often leveraged domain knowledge such as transition structure [Balaji et al., 2020], hierarchies

of sub-actions [Geißer et al., 2020], and known environment model [Chitnis et al., 2021]. However, such prior information is often unavailable, and the true environment model is inaccessible in many domains (e.g., healthcare). Furthermore, it is unclear whether they can be extended to high-dimensional observations (i.e., pixels).

Our motivation stems from the fact that only some of the sub-actions determine the transition from the current state, making others irrelevant in many cases. For example, certain treatments often disable the influence of other medications for some patients. In the context of MCTS, the exploration of those sub-actions irrelevant to the transition would be redundant. It is worth noting that the significance of each sub-action may vary across different states, e.g., due to the varying physiological mechanisms among patients.

In this work, we propose an action abstraction based on the compositional structure between the state and sub-actions that improves the efficiency of MCTS under the factored action space. Our method identifies such relationships by learning a masked latent dynamics model that employs only sub-actions necessary for prediction, which we call state-conditioned action abstraction. Importantly, it does not rely on the true environment model and learns from raw observations. Furthermore, such compositional structure is learned solely with the reconstruction loss, making it also practical under sparse reward environments. During the tree traversal, our method infers the relevant sub-actions on each node *on-the-fly*, guiding the subsequent action abstraction.

We augment MuZero [Schrittwieser et al., 2020] and demonstrate the improved sample efficiency of our method on environments with expansive combinatorial action space. Detailed analysis of our method shows the effectiveness of state-conditioned action abstraction and illustrates that it successfully captures compositional relationships between the state and sub-actions.

Our contributions are summarized as follows:

- We devise a simple and effective method that learns compositional structures among the state and actions from pixels without a known environment model.

- We propose a state-conditioned action abstraction for improving the efficiency of MCTS under the factored action space that considers only the sub-actions relevant to the transition from the current state.

- We demonstrate the superior sample efficiency of our method compared to vanilla MuZero, which suffers from the vast combinatorial action space.

## 2 PRELIMINARIES

A Markov Decision Process (MDP) [Bellman, 1957, Puterman, 2014] is defined by a tuple $\langle \mathcal{S}, \mathcal{A}, P, R, \gamma \rangle$, where $\mathcal{S}$ is a state space, $\mathcal{A}$ is an action space, $P : \mathcal{S} \times \mathcal{A} \rightarrow \mathcal{S}$

is a transition function, $R : \mathcal{S} \rightarrow \mathbb{R}$ is a reward function, and $\gamma \in [0, 1)$ is a discount factor. We consider a discrete action space and deterministic transition. The goal is to find a policy $\pi : \mathcal{S} \times \mathcal{A} \rightarrow [0, 1]$ that maximizes the expected cumulative reward. In this paper, we consider the factored action space $\mathcal{A} = \mathcal{A}^1 \times \cdots \times \mathcal{A}^n$ where the action $A$ is composed of sub-actions $A^i$, i.e., $A = [A^1, \cdots, A^n]$, and each $A^i$ takes values from a set $\mathcal{A}^i$. We will sometimes denote sub-actions as action variables. Throughout the paper, we use capital and small letters to represent the random variables and their assignments, respectively.

### 2.1 MONTE CARLO TREE SEARCH

Monte Carlo Tree Search (MCTS) [Browne et al., 2012, Coulom, 2006] incrementally builds a search tree to find the best decision from a given state. The algorithm's strength lies in its balance between exploring previously underexplored actions and exploiting actions with high estimated rewards. Typically, MCTS iteratively repeats the simulations, which consist of four main stages: *selection*, *expansion*, *evaluation*, and *backup*. Beginning at the root node of the search tree, which represents the current state, MCTS traverses the tree by successively selecting the most promising node until a leaf node is reached.

In our work, we consider a variant of MCTS proposed in MuZero [Schrittwieser et al., 2020], which incorporates policy and value networks with a latent dynamics model to guide tree search effectively when the true environment model is unknown and the agent receives a high-dimensional observation (i.e., pixels). We now briefly describe each model component and the search procedure of MuZero.

**Model components.** The encoder $f$ embeds each state $s_t$ into a latent space, i.e., $z_t = f(s_t)$. Here, the state could be a high-dimensional observation, such as an image. The latent dynamics model $g$ maps the latent state $z_t$ and an action $a_t$ into a next latent state, i.e., $\hat{z}_{t+1} = g(z_t, a_t)$. At each time step $t$, it builds a search tree starting from $z_t$ as a root node and recursively selects an action for each node.

**Selection.** At each node $z$, the action selection is:

$$\hat{a} = \arg\max_a \left[ Q(z, a) + c \cdot \pi_\theta(z, a) \frac{\sqrt{\sum_b N(z, b)}}{1 + N(z, a)} \right], \quad (1)$$

where the estimated Q-value, policy prior, and visit count are denoted as $Q(z, a)$, $\pi_\theta(z, a)$, and $N(z, a)$, respectively. Here, $c = c_1 + \log\left(\frac{\sum_b N(z,b) + c_2 + 1}{c_2}\right)$ is an exploration coefficient where $c_1$ and $c_2$ are hyperparameters and $\sum_b N(z, b)$ represents the total number of visits for all actions from state $z$. The learnable policy prior $\pi_\theta(z, a)$ guides the search towards promising actions.

**Expansion.** If there is no child node corresponding to the selected action during the tree traversal, the latent dynamics

model predicts the subsequent latent state and adds it to the search tree as a child node of the current node.

**Evaluation.** After expanding the search tree, it evaluates a reward and a value from the expanded node. Also, a policy prior is estimated for later use in the *selection* stage.

**Backup.** At the end of a simulation, it updates the visit count and Q-value estimation of the selected nodes along the path from the root to the expanded node. Each $Q(z, a)$ is updated based on the value of the expanded node and the cumulative rewards along the path to the expanded node.

**Training.** The latent dynamics model, encoder, reward, policy, and value networks are jointly trained to predict the policy, value, and reward targets. Specifically, it encodes the state $s_t$ into $z_t$ and unrolls the dynamics model, constructing $\hat{z}_{t+1}, \cdots, \hat{z}_{t+k}$. It then predicts the policy, value, and rewards on each $\hat{z}_{t+i}$. They are supervised to estimate the bootstrapped value, the reward, and the visit count distribution which is the normalized number of visits for each action from the MCTS over the states $s_{t+1}, \ldots, s_{t+k}$.

## 2.2 CONTEXT-SPECIFIC INDEPENDENCE

Our main goal is to improve the efficiency of MCTS in environments with a factored action space. The key motivation is that some of the action variables do not influence the transition in the current state. The notion of context-specific independence (CSI) [Boutilier et al., 2013] provides a way to understand such relationships.

**Definition 2.1** (Context-Specific Independence). We say $Y$ is *contextually independent* of $W$ given the context $X = x$ if $p(y \mid x, v, w) = p(y \mid x, v)$ holds for all $y \in \mathcal{Y}$ and $(v, w) \in \mathcal{V} \times \mathcal{W}$ whenever $p(x, v, w) > 0$. This is denoted by $Y \perp\!\!\!\perp W \mid X = x, V$.

We are concerned with CSI relationship between the current state $s$ and action variables $A = [A^1, \cdots A^n]$, written as:

$$S' \perp\!\!\!\perp A \setminus A_M \mid S = s, A_M,$$

where $M = \{j_1, \cdots, j_m\} \subseteq [n]$, $A_M = [A^{j_1}, \cdots, A^{j_m}]$. Note that this only holds in the current state $s$ and does not generally hold. In other words, sub-actions that influence the state transition may vary across different states.

Existing approaches to capture such compositional structures between the state and sub-actions often rely on *true* environment model [Chitnis et al., 2021], e.g., using conditional independence tests. However, it is impractical for a high-dimensional observation, e.g., image, and more importantly, it is unavailable in many scenarios, e.g., healthcare.

## 3 METHOD

In this section, we describe each component of our method in detail. Overall framework of our method is illustrated in Fig. 2. We first describe a state-conditioned action abstraction, a set of sub-actions relevant to the transition on the current state, and an auxiliary network that infers such relationships (Sec. 3.1). We then describe the training of the latent dynamics model with the auxiliary network (Sec. 3.2), as depicted in Fig. 2(a). Finally, we combine MCTS with state-conditioned action abstraction (Sec. 3.3), as depicted in Fig. 2(b).

## 3.1 STATE-CONDITIONED ACTION ABSTRACTION

As described earlier, our method learns compositional structure between the state and action variables so as to reduce the search space of MCTS. For this, we devise a conditional structure inference network that infers action variables irrelevant of the transition from the current state. Importantly, it operates in the latent space to deal with high-dimensional observations.

First, an encoder $f$ maps the observation (i.e., image) to the latent state representation, i.e., $z = f(s)$. The conditional structure inference network $h$ then infers from $z$ as:

$$h(z) = [p_z^1, \cdots, p_z^n] \in [0, 1]^n, \tag{2}$$

where each entry $p_z^i$ is the parameter of the Bernoulli distribution. The mask is then sampled from $h(z)$ as:

$$M(z) = [m_z^1, \cdots, m_z^n] \in \{0, 1\}^n, \tag{3}$$

where $m_z^i \sim \text{Bernoulli}(p_z^i)$ for all $i \in [n]$. Here, the action variable $A^i$ is relevant for the state transition if $m_z^i = 1$; otherwise, it is irrelevant. Based on this, we construct a state-conditioned action abstraction which is defined as:

$$\phi_z(A) = \{A^i \mid m_z^i = 1\} \subseteq A. \tag{4}$$

It is worth keeping in mind that the abstraction depends on the current state. This represents the CSI relationship as:

$$S' \perp\!\!\!\perp \phi_z^c(A) \mid S = s, \phi_z(A), \tag{5}$$

where $\phi_z^c(A) \coloneqq A \setminus \phi_z(A)$. For example, in the case of 3 action variables $A = [A^1, A^2, A^3]$ with the inferred mask $M(z) = [1, 1, 0]$, the inferred CSI is $S' \perp\!\!\!\perp A^3 \mid S = s, \{A^1, A^2\}$ and the abstract action is $\phi_z(A) = [A^1, A^2]$. We denote $\phi_z(\mathcal{A})$ as the abstract action space, e.g., $\phi_z(\mathcal{A}) = \mathcal{A}^1 \times \mathcal{A}^2$, and denote $\phi_z(a)$ as the value of $\phi_z(A)$, e.g., if $a = [a^1, a^2, a^3]$, then $\phi_z(a) = [a^1, a^2] \in \phi_z(\mathcal{A})$.

Such auxiliary network is utilized to uncover CSI relations when true variables are fully observable in low-dimension [Hwang et al., 2023]. However, it is unclear how to capture

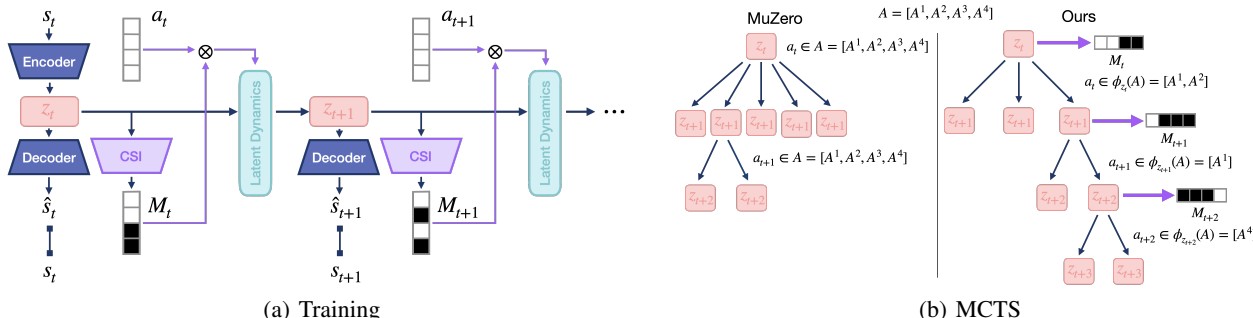

|            |            |
|:----------:|:----------:|
| (a) Training | (b) MCTS |

Figure 2: Overall framework. (a) Training latent dynamics model with conditional structure inference network. (b) The proposed MCTS with state-conditioned action abstraction.

them from high-dimensional observation. We proceed to describe how to learn the conditional structure inference network $h$ that induces the state-conditioned action abstraction $\phi_z(A)$ which adheres to Eq. (5).

### 3.2 TRAINING LATENT DYNAMICS MODEL

A CSI relationship in Eq. (5) implies that the abstract action $\phi_z(a)$ is sufficient for predicting the future state:

$$p(s' \mid s, a) = p(s' \mid s, \phi_z(a)). \tag{6}$$

Thus, we train the latent dynamics model $g$ to use the abstraction action for prediction, i.e., $\hat{z}_{t+1} = g(z_t, \phi_{z_t}(a_t))$. We employ $K$-step reconstruction loss to jointly train the latent dynamics model and conditional structure inference network as Fig. 2(a):

$$\mathcal{L}_{recon}(s_t) = \frac{1}{K} \sum_{k=1}^{K} \Big[ \| s_{t+k} - \mathtt{Dec}(\hat{z}_{t+k}) \|_2^2$$
$$+ \lambda \| M(\hat{z}_{t+k-1}) \|_1 \Big], \tag{7}$$

where $\hat{z}_{t+k} = g(\hat{z}_{t+k-1}, \phi_{\hat{z}_{t+k-1}}(a_{t+k-1}))$ and $\hat{z}_t = z_t = f(s_t)$. $\lambda$ is a sparsity coefficient, which is a hyperparameter. Intuitively, the regularized reconstruction loss encourages the models to accurately predict the future state by using only necessary action variables, i.e., $\phi_z(a)$. This allows us to learn the compositional relationships between the current state and action variables from high-dimensional observations without knowing the true environment model.

Since $M(z) = [m_z^1, \cdots, m_z^n]$ is not differentiable with respect to $z$ due to the sampling $m_z^i \sim \text{Bernoulli}(p_z^i)$, we use Straight-Through Gumbel-Softmax estimator [Maddison et al., 2016, Jang et al., 2016]:

$$\sigma \left( \frac{1}{\beta} (\log p_z^i - \log(1 - p_z^i) + \log u - \log(1 - u)) \right),$$

where $\sigma$ is the sigmoid function, $u \sim \text{Unif}(0,1)$, and $\beta$ is a temperature. Intuitively, $h(z) = [p_z^1, \cdots, p_z^n]$ is trained to

assign a high probability to the sub-action that is necessary for predicting the future state. This allows us to update the conditional structure inference network $h$ with the reconstruction loss with regularization in Eq. (7).

### 3.3 COMPLETE METHOD: MCTS WITH STATE-CONDITIONED ACTION ABSTRACTION

We propose MCTS using abstract action $\phi_z(a)$ for each node $z$, instead of $a$, reducing the search space exponentially with respect to the number of sub-actions masked out. An overall framework is illustrated in Fig. 2(b).

**Deterministic abstraction.** State-conditioned action abstraction (Eq. (4)) involves the sampling from a Bernoulli distribution. For the inference, we use a deterministic abstraction with the threshold $\tau$, which is a hyperparameter:

$$\phi_z(A) = \{A^i \mid p_z^i > \tau\} \subseteq A. \tag{8}$$

**Selection.** At each node $z$, an abstract action is selected as:

$$\widehat{\phi}_z(a) = \arg\max_{\phi_z(a)} \Big[ Q(z, \phi_z(a)) + \tag{9}$$
$$c \cdot \pi_\theta(z, \phi_z(a)) \frac{\sqrt{\sum_b N(z, b)}}{1 + N(z, \phi_z(a))} \Big],$$

where the abstract action $\phi_z(a)$ is the key difference compared to the vanilla action selection of MuZero in Eq. (1). Here, the policy prior $\pi_\theta(z, a)$ is marginalized over the actions $a'$ that are projected to the same abstract action $\phi_z(a)$:

$$\pi_\theta(z, \phi_z(a)) = \sum_{\{b \in \mathcal{A} \mid \phi_z(b) = \phi_z(a)\}} \pi_\theta(z, b). \tag{10}$$

For example, if $A = [A^1, A^2, A^3]$ where the action variables are binary, $\phi_z(A) = [A^1, A^2]$, and $\phi_z(a) = (0,0)$, then we are marginalizing over the third dimension: $\pi_\theta(z, \phi_z(a)) = \pi_\theta(z, (0,0,0)) + \pi_\theta(z, (0,0,1))$. Note that the modeling of the policy prior as $\pi_\theta(z, a)$ instead of $\pi_\theta(z, \phi_z(a))$ is the

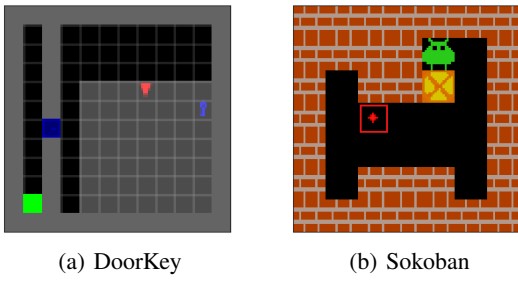

(a) DoorKey      (b) Sokoban

Figure 3: Sample images for each environment.

design choice for simplicity since the abstraction depends on the current state, making the dimension of $\phi_z(a)$ varies across different states.

**Expansion and backup.** If there is no child node corresponding to the selected action $\widehat{\phi}_z(a)$, the latent dynamics model predicts the subsequent latent state $z' = g(z, \widehat{\phi}_z(a))$ and adds it to the search tree as a child node of the current node $z$. The rest of the procedures are identical to MuZero.

**Final action selection at the root node.** After the simulations, the final action is selected based on the visit distribution $\hat{\pi}(z, \phi_z(a))$, which is the normalized visit count for each (abstract) action from the root node $z$. We unfold the visit distribution to the original action space $A$ as:

$$\hat{\pi}(z, a) = \hat{\pi}(z, \phi_z(a)) \times u(\phi_z^c(a)), \qquad (11)$$

where $u(\phi_z^c(a))$ represents the uniform distribution over action variables $\phi_z^c(a)$. This provides diverse state-action samples for robust training of the auxiliary network.

**Training.** All components of our method are jointly trained in an end-to-end fashion. The conditional structure inference network is trained only with the reconstruction loss to faithfully represent the dynamics transition in Eq. (6). The remaining components are trained with the combination of policy, value, reward, and reconstruction losses, similar to MuZero as described in Sec. 2.1.

# 4 EXPERIMENTS

In this section, we evaluate our method on environments with expansive combinatorial action spaces. Our investigation focuses on (1) whether the proposed MCTS with state-conditioned action abstraction improves the sample efficiency of vanilla MuZero (Figs. 4, 5 and 10), (2) whether our method successfully captures compositional relationships between the state and sub-actions (Figs. 6 to 8 and 12), and (3) how much the action abstraction contributes to the sample efficiency (Figs. 9 and 11 and table 1).[1]

---

[1]Our code is available at https://github.com/yun-kwak/efficient-mcts.

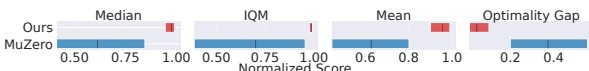

Figure 4: Comparison of aggregate metrics across all tasks.

## 4.1 EXPERIMENTAL SETUP

Following the implementation [Schrittwieser et al., 2020, 2021], we augment MuZero by incorporating the proposed conditional structure inference network. We use vanilla MuZero as a baseline throughout the experiments.

**Implementation.** We use the abstraction threshold $\tau = 0.01$ for all experiments. For all environments, we train our method and MuZero over 100k gradient steps and evaluate the performance of each run for every 2000 steps with 32 seeds. All experiments were executed on an NVIDIA RTX 3090 GPU, leveraging JAX and Haiku. We provide implementation details in Appendix B. Additional experimental results are provided in Appendix C.

### 4.1.1 Environments

**DoorKey** [Chevalier-Boisvert et al., 2018]. We modify the MiniGrid DoorKey environment to introduce a factored action space (Fig. 3(a)). The task is to obtain a key, open a door, and ultimately reach a designated goal, following the shortest path. The action space is factorized as $\mathcal{A} = \mathcal{A}_{\text{turn}} \times \mathcal{A}_{\text{forward}} \times \mathcal{A}_{\text{pick}} \times \mathcal{A}_{\text{open}}$, where $\{\mathcal{A}_{\text{turn}}, \mathcal{A}_{\text{forward}}\}$ correspond to the movement of the agent. $\mathcal{A}_{\text{pick}}$ and $\mathcal{A}_{\text{open}}$ correspond to the interaction with the key and door, respectively. The agent receives a reward of $-0.1$ for each step taken. The configuration of the door, key, wall, and the initial position of the agent is randomly initialized at the beginning of each episode. The attributes of the door and the key are also randomly initialized in each episode. We design three settings: EASY, NORMAL, and HARD, where the number of the attributes is 2, 3, and 4, respectively. The action cardinality for each setting is 54, 96, and 150. Details of the DoorKey are provided in Appendix A.1.

**Sokoban** [Schrader, 2018]. It is a challenging environment that requires long-horizon planning where the agent must manipulate a box to a designated target location through a series of actions (Fig. 3(b)). The map topology, goal location, and box attribute are randomly initialized for each episode. The agent receives a reward of $-0.1$ for each step taken and a reward of 10 for successfully placing the box on the target location. The action space is factorized as $\mathcal{A} = \mathcal{A}_{\text{move}} \times \prod_i \mathcal{A}_{\text{box}}^{(i)}$, where each $\mathcal{A}_{\text{box}}^{(i)}$ represents the manipulation of the corresponding box. Similar to DoorKey, we design three settings: EASY, NORMAL, and HARD, where the number of the attributes of the box is 2, 3, and 4, respectively. The action cardinality for each setting is 45, 135, and 405. Details of the Sokoban are provided in Appendix A.2.

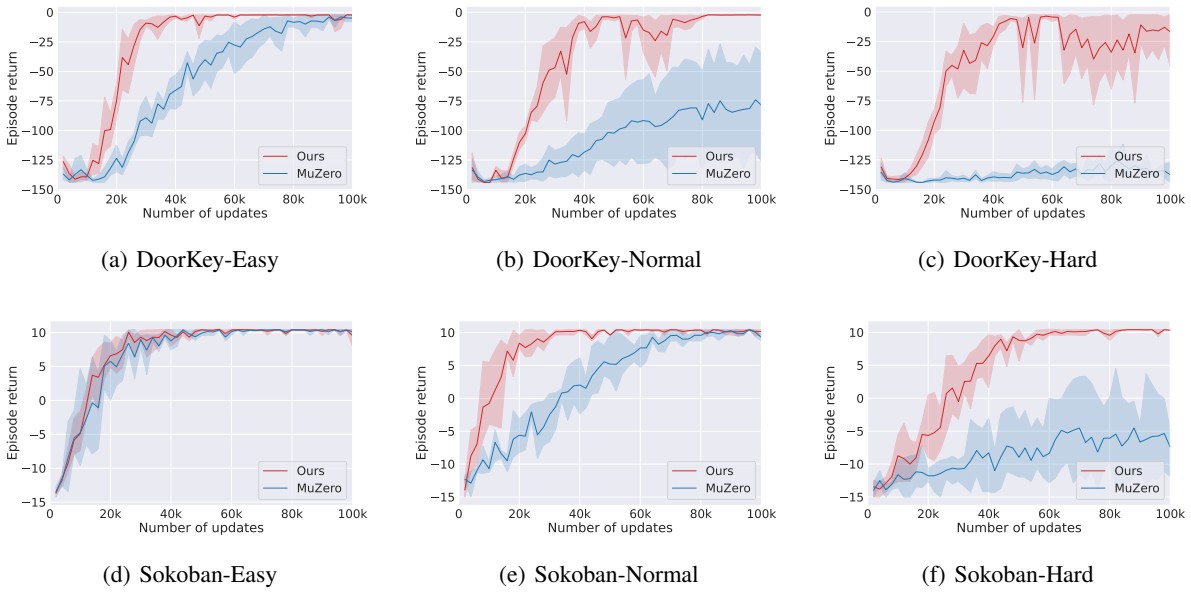

Figure 5: Learning curves. The average episodic return is depicted by lines, and the shaded areas represent the 95% confidence intervals.

## 4.2 RESULTS

**Sample efficiency (Fig. 5).** We measure the episodic returns of our method and MuZero in each environment. While MuZero struggles with vast combinatorial action space, our method achieves near-optimal performance in all environments. We observe that the gap between our method and MuZero becomes more pronounced as it gets harder, i.e., as the action cardinality increases. In particular, our method successfully solves HARD settings, where the action cardinality is 150 for DoorKey and 405 for Sokoban. Following the suggestions from Agarwal et al. [2021], we also report aggregate scores across all runs on DoorKey and Sokoban environments in Fig. 4. The normalized scores for each environment are shown in Fig. 1, illustrating that our method remains effective under the expansive action space.

**Visualization of the state-conditioned action abstraction (Fig. 6).** The conditional structure inference network $h$ learns CSI relationships between state and action variables. We visualize the output $h(z) = [p_z^1, \cdots, p_z^n] \in [0,1]^n$ (Eq. (2)) on different states in DoorKey. We first recall that the action is factorized as $\mathcal{A} = \mathcal{A}_{\text{turn}} \times \mathcal{A}_{\text{forward}} \times \mathcal{A}_{\text{pick}} \times \mathcal{A}_{\text{open}}$ and the agent always turns first, advances, and then either picks up a key or opens a door. Fig. 6-(a) shows that the sub-actions corresponding to the key and the door are assigned almost zero probability. This is because the agent has already obtained a key and cannot interact with the door at this moment. In Fig. 6-(b), the agent is able to pick up the key, and thus, the probability close to 1 is assigned to the sub-action corresponding to the key. Since it still cannot interact with the door, our method accurately predicts the

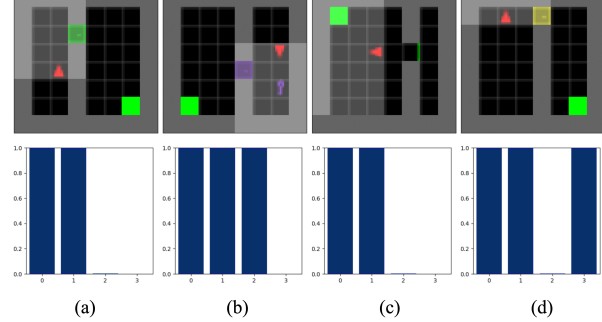

Figure 6: Visualization of state-conditioned action abstraction. (**Top**) observations. (**Bottom**) the probability of dependencies for each action variable.

corresponding probability close to 0. Similarly, Fig. 6-(c) is also the case where the agent has already obtained a key and opened the door, and consequently, the corresponding sub-actions become irrelevant. Finally, in Fig. 6-(d), the agent is able to interact with the door. Our model assigns the probability of 0 and 1 to the sub-action corresponding to the key and the door, respectively.

**Evaluation of CSI discovery (Fig. 7).** We evaluate the performance of the conditional structure inference network using the Structural Hamming Distance (SHD) [Acid and de Campos, 2003, Ramsey et al., 2006], which measures the difference between two directed graphs. A lower SHD indicates that the two graphs are more similar in structure, whereas an SHD of 0 means the graphs are identical. As the environments in our main experiments (i.e., DoorKey

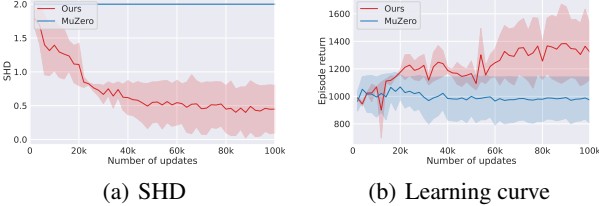

(a) SHD        (b) Learning curve

Figure 7: Evaluation of CSI identification and learning curves on contextual multi-armed bandit.

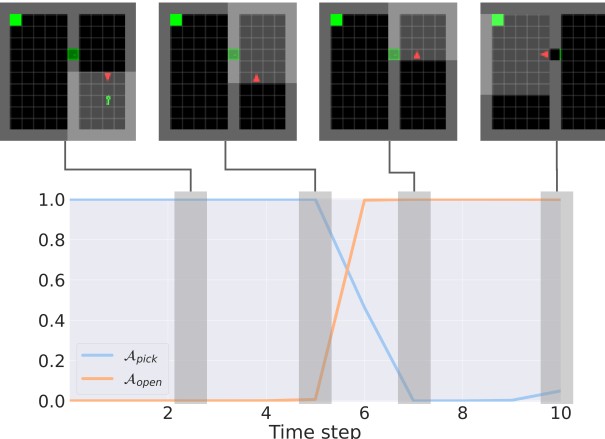

Figure 8: Prediction of the conditional structure inference network for the sub-actions corresponding to the key ($\mathcal{A}_{\text{pick}}$) and the door ($\mathcal{A}_{\text{open}}$) along the trajectory in DoorKey.

and Sokoban) do not provide the ground truth CSIs, we designed the contextual multi-armed bandit scenario for the evaluation with SHD. As shown in Fig. 7(a), our method successfully identifies CSI relationships. Furthermore, Fig. 7(b) illustrates that the performance of our method measured with the episodic return becomes more improved as it becomes more accurately identifies CSIs. This demonstrates the importance of capturing compositional structures and state-conditioned action abstraction for our method. We provide additional details in Appendix A.3.

**Conditional structure inference network** $h$ **(Fig. 8).** We further investigate the behavior of the conditional structure inference network by examining how its inference on the compositional relationships changes along the trajectory. Interestingly, when the agent is close to the key but has not picked it up yet ($t = 2$), our method (Eq. (2)) infers $p_z^{\text{pick}} = 1$ and $p_z^{\text{open}} = 0$. This is because it cannot open the door at the moment, and thus, the corresponding sub-action $A^{\text{open}}$ is irrelevant. Then, it proceeds to obtain the key and starts to move toward the door ($t = 5$). From this moment, the prediction of the conditional structure inference network begins to change. When the agent comes close to the door ($t = 7$) and finally opens it ($t = 10$), our method predicts $p_z^{\text{pick}} = 0$ and $p_z^{\text{open}} = 1$. This illustrates that our

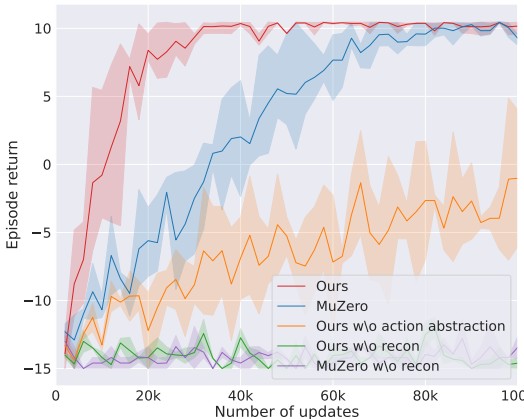

Figure 9: Ablations on action abstraction and reconstruction loss in Sokoban-Normal.

method effectively captures the compositional relationships between the current state and action variables that change across different states and time-steps.

**Ablation study (Fig. 9).** It is clear that the reconstruction loss is crucial for both our method and MuZero. In addition, the performance improvement of using the state-conditioned action abstraction illustrates that it significantly contributes to the superior sample efficiency of our method. In fact, our method without action abstraction performs similar to MuZero, indicating that masked latent dynamics modeling alone does not bring any performance gain.

**Simulation budgets (Fig. 10).** We compare our method with MuZero in DoorKey across a varying number of simulations. Our method consistently outperforms MuZero and achieves near-optimal performance in all budgets. These results illustrate the effectiveness of state-conditioned action abstraction guided by conditional structure inference network, leading to superior sample efficiency and scalability.

**Latent dynamics model (Table 1 and fig. 11).** We examine the latent dynamics model to investigate whether the superior sample efficiency of our method comes from the dynamics model or state-conditioned action abstraction. As shown in Table 1, the latent dynamics model of MuZero achieves slightly better performance in terms of the reconstruction loss. This is because our method imposes the dynamics model to use only some of the action variables for prediction. This illustrates that the superior performance of our method does not come from the latent dynamics model but state-conditioned action abstraction. We further investigate the reconstructed observations from MuZero and ours. As shown in Fig. 11, both dynamics models perform reasonably well, demonstrating the effectiveness of the state-conditioned action abstraction.

**GradCAM (Fig. 12).** We visualize the learned conditional structure inference network using GradCAM [Selvaraju

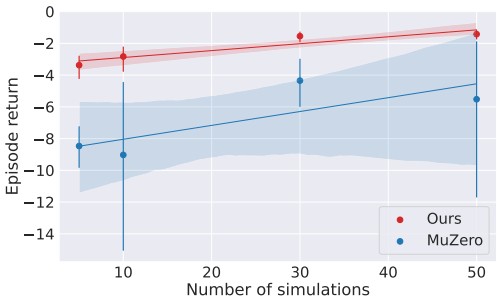

Figure 10: Episodic return per number of simulations.

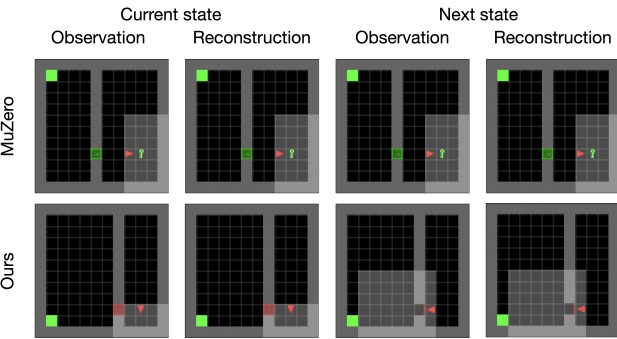

Figure 11: Examples of the observation reconstructions.

et al., 2016] on DoorKey. We use the gradient of $p_z^i$ for the sub-action $A^i$ with respect to the last feature map. We observe that our method places attention on the related object to decide whether the corresponding sub-action is relevant or not. For example, in Fig. 12(a), the network focuses on the placed key and the position of the agent to determine the probability assigned to the sub-action $A^{\text{key}}$ corresponding to the key. Similarly, Fig. 12(b) shows that our model focuses on the door to infer the state-conditioned dependency of the sub-action $A^{\text{door}}$ corresponding to the door.

## 5  RELATED WORK

**Improving the efficiency of MCTS under expansive action space.** There are numerous works on MCTS utilizing known environment models [Hostetler et al., 2014, Xu et al., 2022, Silver et al., 2017, 2016, Couëtoux et al., 2011, Kim et al., 2020]. MuZero [Schrittwieser et al., 2020] incorporates dynamics learning into MCTS, demonstrating its capability to solve complex sequential decision making tasks even when the true model is unavailable. There also exist several variants of MuZero [Grill et al., 2020, Ozair et al., 2021], e.g., which are further extended to handle stochastic transition [Sokota et al., 2021]. As the number of actions or potential outcomes for each action increases, the branching factor expands, posing challenges when facing vast combinatorial action space. Consequently, several studies [Hoerger et al., 2023, Chitnis et al., 2021] have focused on reducing the branching factor in MCTS by employing

Table 1: Reconstruction Loss.

| Models | DoorKey | | | Sokoban | | |
|---|---|---|---|---|---|---|
| | Easy | Normal | Hard | Easy | Normal | Hard |
| MuZero | $0.21_{\pm 0.14}$ | $0.05_{\pm 0.03}$ | $0.08_{\pm 0.10}$ | $0.02_{\pm 0.00}$ | $0.05_{\pm 0.01}$ | $0.04_{\pm 0.01}$ |
| Ours | $0.85_{\pm 0.43}$ | $0.94_{\pm 0.62}$ | $0.70_{\pm 0.48}$ | $0.02_{\pm 0.00}$ | $0.03_{\pm 0.02}$ | $0.03_{\pm 0.01}$ |

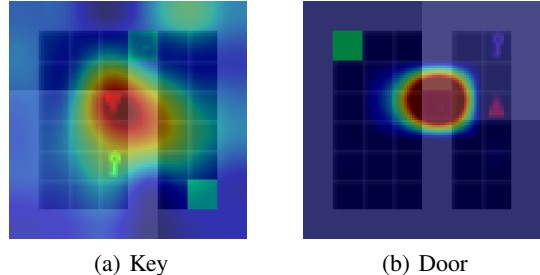

(a) Key        (b) Door

Figure 12: GradCAM visualization.

state or action abstraction techniques. Similar to our work, Chitnis et al. [2021] selects the most useful CSI relationship for a given task and constructs state abstraction with it. However, they do not modify the inside mechanism of MCTS, which limits its applications since its abstraction is fixed throughout the whole episode. Moreover, they require a known environment model to identify such relationships, which is impractical in many scenarios involving an unknown model and high-dimensional observations. In contrast, our method constructs action abstraction for each node *on-the-fly*, leading to more efficient tree traversal through flexible abstraction (Fig. 2(b)). Furthermore, the auxiliary network identifies such relationships from pixels without a known environment model, highlighting its practicality.

**MCTS under factored action space.** Broadly, several studies [Tang et al., 2022, Rebello et al., 2023, Tkachuk et al., 2023, Mahajan et al., 2021] demonstrated the benefits of utilizing the factorized structure in MDP. In the context of MCTS, Geißer et al. [2020] leveraged factored action space by building subtrees for each action variable. However, their hierarchical order significantly influences the algorithm, and thus, the prior information on the relationships among the action variables is crucial. Balaji et al. [2020] proposed to learn the dynamics model with the factored graphs representing the conditional independencies between state and action variables, which is assumed to be known. In contrast, our method does not rely on such domain knowledge and effectively extracts the compositional structure of the state and action variables along with the training of the latent dynamics model.

## 6  CONCLUSION

In this paper, we proposed a novel approach to extending MCTS under factored action spaces that addresses the chal-

lenges posed by large combinatorial action spaces. The proposed method identifies compositional structures between the state and action variables from high-dimensional observations without the true environment model. Based on this, our method constructs state-conditioned abstraction for each node in an on-the-fly manner during the tree traversal. Experimental results demonstrate that our approach significantly improves the sample efficiency of vanilla MCTS under the factored action spaces.

One of the promising future directions to extend our approach is to combine it with state abstraction methods such as bisimulation [Larsen and Skou, 1989, Ferns and Precup, 2014, Zhang et al., 2020]. Recall that our method is about action abstraction for efficient MCTS, such approaches are orthogonal to our work, and we expect that they can be seamlessly integrated into our approach. For example, it would be possible to apply MCTS with our proposed state-conditioned action abstraction on the latent *abstract* state, which we defer to future work.

## Acknowledgements

We would like to thank anonymous reviewers for constructive feedback. This work was partly supported by the IITP (RS-2021-II212068-AIHub/10%, RS-2021-II211343-GSAI/10%, 2022-0-00951-LBA/10%, 2022-0-00953-PICA/20%), NRF (RS-2023-00211904/20%, RS-2023-00274280/10%, RS-2024-00353991/10%), and KEIT (RS-2024-00423940/10%) grant funded by the Korean government.

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

# Efficient Monte Carlo Tree Search via On-the-Fly State-Conditioned Action Abstraction
## (Supplementary Material)

**Yunhyeok Kwak**[†1]     **Inwoo Hwang**[*1]     **Dooyoung Kim**[1]     **Sanghack Lee**[‡1,2]     **Byoung-Tak Zhang**[†1]

[1] AI Institute, Seoul National University
[2] Graduate School of Data Science, Seoul National University

## A  ENVIRONMENTAL DETAILS

For the DoorKey and Sokoban environments, the top-down view of the map is given as the image observation to the agent.

### A.1  DOORKEY

We modify the MiniGrid DoorKey environment [Chevalier-Boisvert et al., 2018] to incorporate a factored action space. The task is first to obtain a key, open a door, and finally reach a goal position. At the initial state, the door is locked, and the agent is positioned on the opposite side of the goal. A wall with a door embedded in it separates the initial position and the goal. The action space is factorized as $\mathcal{A} = \mathcal{A}_{\text{turn}} \times \mathcal{A}_{\text{forward}} \times \mathcal{A}_{\text{pick}} \times \mathcal{A}_{\text{open}}$, where $\mathcal{A}_{\text{turn}} = \{\text{no-op}, \text{turn left}, \text{turn right}\}$, $\mathcal{A}_{\text{forward}} = \{\text{no-op}, \text{move forward}\}$, $\mathcal{A}_{\text{pick}} = \{\text{no-op}, \text{pick red key}, \text{pick blue key}, \dots\}$, and $\mathcal{A}_{\text{open}} = \{\text{no-op}, \text{open red door}, \text{open blue door}, \dots\}$. The "no-op" action denotes a choice to perform no operation, i.e., "do nothing", for that particular action set. If the number of colors is 4, the cardinality of the action space is $|\mathcal{A}| = 3 \times 2 \times 5 \times 5 = 150$. We introduce difficulty levels (EASY, NORMAL, HARD) corresponding to two, three, and four colors, respectively. The action cardinality for each setting is 54, 96, and 150. The agent always turns first, then progresses by advancing, and finally completes the action by retrieving a key and/or opening a door. The horizon $H$ is 1440, and the agent receives a reward of $-0.1$ for each step. The configurations of the door, key, wall, and the initial position of the agent are randomly initialized at the beginning of each episode. We conducted experiments with a $12 \times 12$ size for primary results (Figs. 4 and 5) and an $8 \times 8$ size for supplementary studies (Figs. 6, 10 and 12).

### A.2  SOKOBAN

Sokoban [Chevalier-Boisvert et al., 2018] is a challenging environment that requires long-horizon planning. Here, the task is to move a box to a designated target location through a series of actions. We modify the environment to incorporate a factored action space and generate image observations using the tiny world rendering mode with a handful of visual complexity. The influence exerted on the box is determined solely by a single box-action variable, which depends on the color of the box. In addition, the map configuration, goal location, and box color attribute are randomly initialized at the beginning of each episode. With a horizon of $H = 150$, the agent receives a reward of $-0.1$ for each step taken and a reward of $+10$ for successfully placing the box on the target location. The action space is factorized as $\mathcal{A} = \mathcal{A}_{\text{move}} \times \mathcal{A}_{\text{box}}^{\text{red}} \times \mathcal{A}_{\text{box}}^{\text{blue}} \times \cdots$, where $\mathcal{A}_{\text{move}} = \{\text{no-op}, \text{move up}, \text{move down}, \text{move left}, \text{move right}\}$ and $\mathcal{A}_{\text{box}}^i = \{\text{no-op}, \text{push}, \text{pull}\}$ for all $i \in \{\text{red}, \text{blue}, \dots\}$. If the number of colors is 4, the cardinality of the action space is $|\mathcal{A}| = 5 \times 3^4 = 405$. We introduce difficulty levels (EASY,

---

[*] Equal contribution.
[†] Corresponding authors.
[†] Equal contribution.
[‡] Corresponding authors.

NORMAL, HARD) corresponding to two, three, and four colors, respectively. The action cardinality for each setting is 45, 135, and 405.

## A.3 CONTEXTUAL MULTI-ARMED BANDIT

Our method aims to identify CSI relationships. To rigorously test its performance in capturing CSIs, we employ a Contextual Multi-Armed Bandit (CMAB) environment. We quantify the similarity between the ground truth CSIs and identified CSIs with the Structural Hamming Distance (SHD) [Acid and de Campos, 2003]. The CMAB problem introduces contextual information to the classical Multi-Armed Bandit problem, allowing the agent to leverage state-specific cues for action selection and maximize cumulative rewards over time. The synthetic environment features a combinatorial action space [Chen et al., 2016, Qin et al., 2014, Chen et al., 2018], factorized as $\mathcal{A} = \mathcal{A}^1 \times \mathcal{A}^2 \times \mathcal{A}^3$ where $|\mathcal{A}^i| = 7$. The reward received at each time step is equivalent to the current state $r_t = s_t$. With a horizon of $H = 25$, state space $\mathbb{N}_0$, and $s_0 = 0$, state transitions are solely determined by a sub-action, which varies across states. The sub-action is identified by the ground truth CSI $M_t = \{i\}$, where $i = \mathrm{mod}(\lfloor \frac{s_t}{6} \rfloor, 3)$, and thus this variable is state-dependent. Formally, the transition is $s_{t+1} = s_t + a_t^i$ for even $s_t$, otherwise $s_{t+1} = s_t + (6 - a_t^i)$, where $i$ indicates the index of relevant action variable for state $s_t$. Identifying the relevant action variable for a given state is essential due to the large combinatorial action space ($|\mathcal{A}| = 7^3 = 343$).

# B IMPLEMENTATION DETAILS

## B.1 EXPERIMENTAL DETAILS

We employed the Adam optimizer [Kingma and Ba, 2015] with decoupled weight decay [Loshchilov and Hutter, 2019] for training. Each run spanned approximately 48 hours across both environments. Our method and MuZero were trained over 100k gradient steps (5 runs for DoorKey, 3 for Sokoban). Results in Fig. 10 are averaged across 7 runs per simulation budget. We use a uniform replay buffer and collect 32k transitions under a uniform random policy prior to training. Periodic evaluations were conducted every 2000 update steps using 32 seeds. The confidence intervals of the reported performances were estimated by the $95\%$ percentile bootstrap. Scores are min-max normalized (-150 to 0 for DoorKey, -15 to 10.5 for Sokoban) in Figs. 1 and 4, and average episodic return at 40k gradient steps is shown in Fig. 1. All experiments were executed on an NVIDIA RTX 3090 GPU, leveraging JAX and Haiku. We perform no data augmentation of the state. The $96 \times 96 \times 3$ state is scaled by $s/255$. The action is factorized and encoded as a one-hot vector per action variable, then one-hot vectors are concatenated into a vector for the action. Adhering to Schrittwieser et al. [2020], the action is spatially tiled and paired with the state for input into the dynamics network. For the CMAB environment, the state was encoded into a $96 \times 96 \times 3$ representation through repetition of the normalized state, which is calculated as: $s/((\max_i |\mathcal{A}^i| - 1) \times H)$. Here, $\max_i |\mathcal{A}^i|$ equals 7 and $H$ is set to 25.

## B.2 IMPLEMENTATION OF MUZERO

We mostly follow the architectural design of MuZero from Schrittwieser et al. [2020, 2021]. Following Schrittwieser et al. [2020], we incorporated categorical representations for the value and reward predictions. Dirichlet noise is added to the policy prior as follows: $(1 - \rho)\pi(a|s) + \rho \mathcal{N}_\mathcal{D}(\xi)$, where $\rho = 0.25$, the $\mathcal{N}_\mathcal{D}(\xi)$ is the Dirichlet noise distribution, and $\xi = 0.0$ during evaluations and for non-root nodes, otherwise the noise ratio is set to 0.3. Similar to Ye et al. [2021], we reduce the number of residual blocks and channel dimensions due to the high computational cost of the original network architecture used in MuZero. We use the kernel size 3×3 for all operations, unless otherwise specified. The architecture comprises four components: the representation network, the dynamics network, the prediction network, and the reconstruction network.

The architecture of **the representation network** is as follows:

- 1 convolution with stride 2 and 32 output planes, output resolution 48x48. (LayerNorm + ReLU)
- 1 residual block with 32 planes.
- 1 residual downsample block with stride 2 and 64 output planes, output resolution 24x24.
- 1 residual block with 64 planes.
- Average pooling with stride 2, output resolution 12x12. (LayerNorm + ReLU)
- 1 residual block with 64 planes.

Table 2: Hyperparameters for MuZero and Ours

| Models | Parameters | DoorKey | Sokoban | CMAB |
|---|---|---|---|---|
| MuZero | Observation down-sampling | 96×96 | 96×96 | 96×96 |
| | Frames stacked | No | No | No |
| | Frames skip | No | No | No |
| | Reward clipping | No | No | No |
| | Discount factor | 0.997 | 0.997 | 0.997 |
| | Minibatch size | 256 | 256 | 256 |
| | Optimizer | Adam | Adam | Adam |
| | Learning rate | 0.001 | 0.001 | 0.001 |
| | Momentum | 0.9 | 0.9 | 0.9 |
| | Weight decay | 1e-4 | 1e-4 | 1e-4 |
| | Max gradient norm | 100 | 5 | 100 |
| | Training steps | 100K | 100K | 100K |
| | Evaluation episodes | 32 | 32 | 32 |
| | Min replay size for sampling | 32K | 32K | 32K |
| | Max replay size | 1M | 1M | 1.6M |
| | Target network updating interval | 200 | 200 | 200 |
| | Unroll steps | 5 | 5 | 5 |
| | TD steps | 5 | 5 | 5 |
| | Policy loss coefficient | 1 | 1 | 1 |
| | Value loss coefficient | 0.25 | 0.25 | 0.25 |
| | Reconstruction loss coefficient | 1 | 0.1 | 1 |
| | Dirichlet noise ratio | 0.3 | 0.3 | 0.3 |
| | Number of simulations in MCTS | 50 | 50 | 15 |
| | Reanalyzed policy ratio | 1.0 | 1.0 | 1.0 |
| Ours | Sparsity coefficient | 0.0 | 0.0 | 0.01 |
| | Gumbel sigmoid temperature | 1.0 | 1.0 | 1.0 |
| | MCTS mask threshold | 0.01 | 0.01 | 0.01 |

- Average pooling with stride 2, output resolution 6x6. (LayerNorm + ReLU)
- 1 residual block with 64 planes.

The architecture of **the dynamics network** is as follows:

- Concatenate the input states and input actions.
- 1 convolution with stride 2 and 64 output planes. (LayerNorm)
- A residual link: add up the output and the input states. (ReLU)
- 1 residual block with 64 planes.

The architecture of **the prediction network for the reward prediction** is as follows:

- 1 1x1 convolution with 16 output planes. (LayerNorm + ReLU)
- Flatten
- 1 fully connected layers and 32 output dimensions (LayerNorm + ReLU)
- 1 fully connected layers and 601 output dimensions.

The architecture of **the prediction network for the policy and value prediction** is as follows:

- 1 residual block with 64 planes.
- 1 1x1 convolution with 16 output planes. (LayerNorm + ReLU)
- Flatten
- 1 fully connected layers and 32 output dimensions. (LayerNorm + ReLU)

- 1 fully connected layers and D output dimensions,

where $D = 601$ in the value prediction network and $D = |\mathcal{A}|$ in the policy prediction network. The policy and value prediction network shares the initial residual block.

The architecture of **the reconstruction network** is as follows:

- 1 residual block with transposed convolution, stride 1, and 64 output planes.
- 1 residual block with transposed convolution, stride 2, and 64 output planes.
- 1 residual block with transposed convolution, stride 1, and 64 output planes.
- 1 residual block with transposed convolution, stride 2, and 64 output planes.
- 1 residual block with transposed convolution, stride 1, and 64 output planes.
- 1 residual block with transposed convolution, stride 2, and 32 output planes.
- 1 residual block with transposed convolution, stride 1, and 32 output planes.
- 1 transposed convolution with stride 2 and 3 output dimensions. (LayerNorm + ReLU)

We use mostly the same hyperparameter setting as presented in Ye et al. [2021]. Details of the hyperparameters are provided in Table 2.

## B.3 IMPLEMENTATION OF OUR METHOD

We build our method on top of the implementation of MuZero. Our method introduces additional hyperparameters required for action abstraction, i.e., the sparsity regularization coefficient $\lambda$, the Gumbel sigmoid temperature $\delta$, and the abstraction threshold $\tau$. We use the abstraction threshold $\tau$ to induce on-the-fly action abstraction from the probabilities of state-conditioned dependencies for each action variable. The sparsity coefficient is set to $\lambda = 0$ on the DoorKey and Sokoban environments, and $\lambda = 0.01$ for the CMAB. We set the abstraction threshold and the Gumbel sigmoid temperature to $\tau = 0.01$ and $\delta = 1$, respectively, for all experiments. For our conditional structure inference network, we use Gumbel-Softmax reparametrization for backpropagation [Maddison et al., 2016, Jang et al., 2016], similar to Hwang et al. [2023].

The architecture of **the auxiliary network for discovering CSIs** is as follows:

- 1 residual block with 64 planes.
- 1 1x1 convolution with 16 output planes. (LayerNorm + ReLU)
- Flatten
- 1 fully connected layers and 32 output dimensions (LayerNorm + ReLU)
- 1 fully connected layers and D output dimensions,

where $D$ is the number of action variables and the initial residual block is shared with the policy and value prediction network.

## C ADDITIONAL EXPERIMENTS

In this section, we present additional visualizations (Figs. 13 and 14) and ablation results (Fig. 15) to underscore the benefits of our approach. Experiments in Fig. 15 used a more complex DoorKey environment (five colors, 216 actions vs. four colors, 150 actions in HARD difficulty). Despite the vast combinatorial action space, our method demonstrates near-optimal performance in Fig. 15, significantly outperforming vanilla MuZero. Ablations in Figs. 15(a) and 15(b) confirm that the state-conditioned action abstraction is a vital part of our improvement.

We further investigate the effect of training the conditional structure inference network only with the reconstruction loss to faithfully represent the dynamics transition, as described in Sec. 3.3. We train the conditional structure inference network only with the gradients from the reconstruction loss to ensure that it learns the proper context-specific independence from the transition dynamics. In other words, we freeze the parameters of the conditional structure inference network when learning policy, value, and reward. Fig. 15(c) shows that updating the network with solely reconstruction loss (Frozen) enhances stability compared to the conditional structure inference network jointly trained with all losses (Unfrozen).

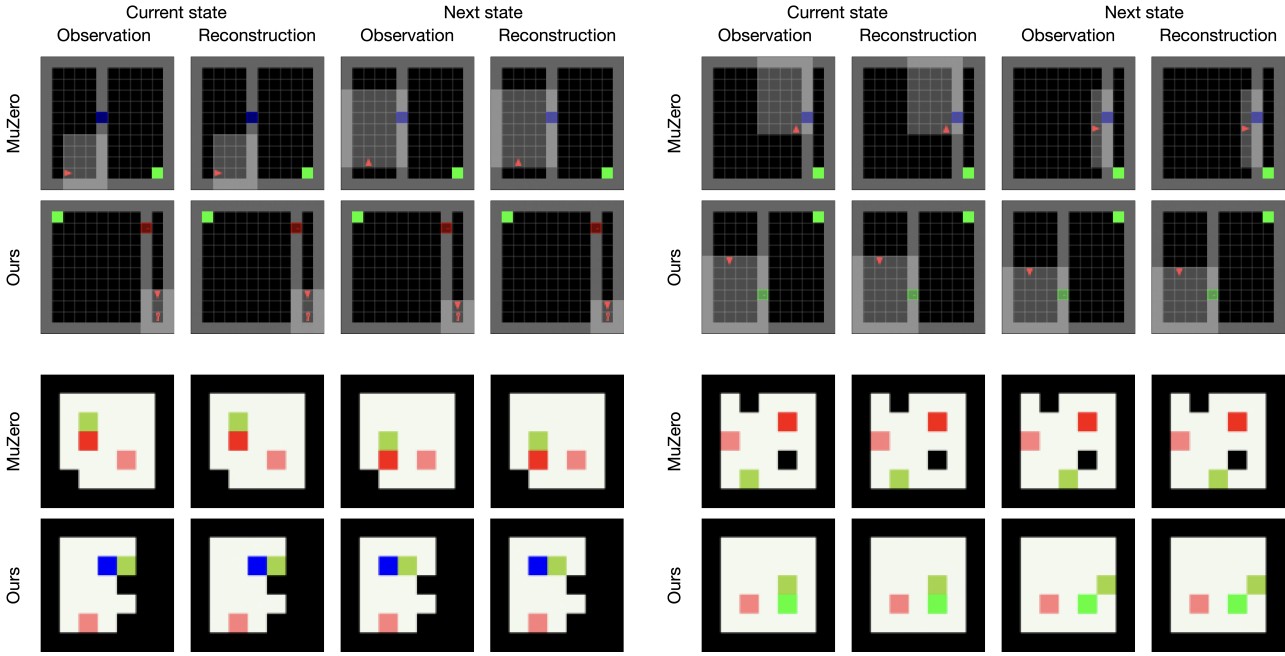

Figure 13: Additional examples of the observation reconstructions.

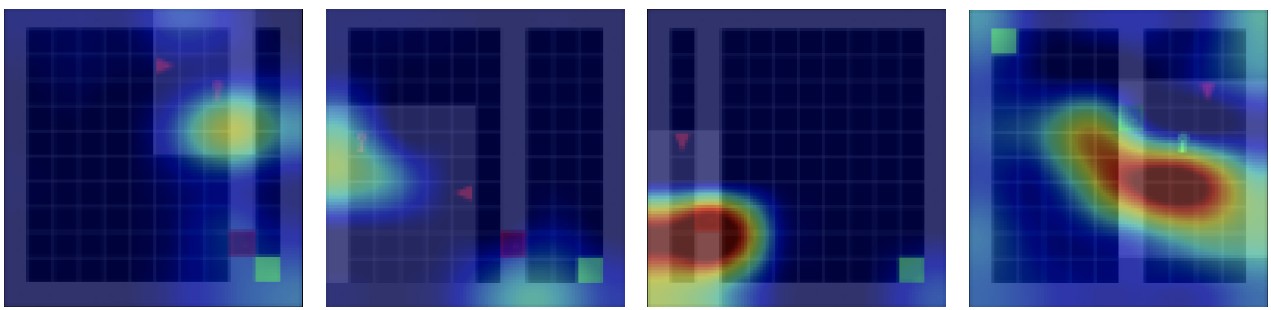

Figure 14: Additional GradCAM visualization.

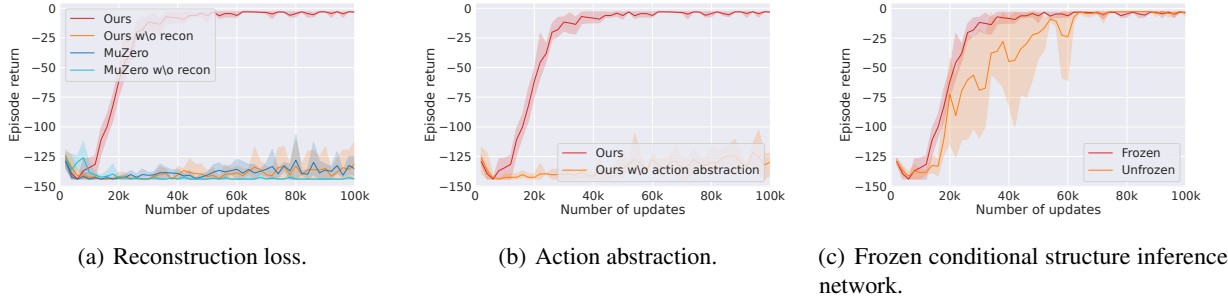

(a) Reconstruction loss.

(b) Action abstraction.

(c) Frozen conditional structure inference network.

Figure 15: Additional results from the ablations. The average episodic return across 3 runs is depicted by lines, and the shaded areas represent the $95\%$ confidence intervals.

**Sparsity coefficient** $\lambda$**.** We report experimental results from an ablation analysis of the sparsity coefficient $\lambda$ in Fig. 16. We evaluated our method with $\lambda$ values of $\{0.0, 0.01, 0.001\}$ on the DoorKey-Easy environment, and the performance of MuZero is also presented for comparison. The results demonstrate that our approach maintains a considerable degree of

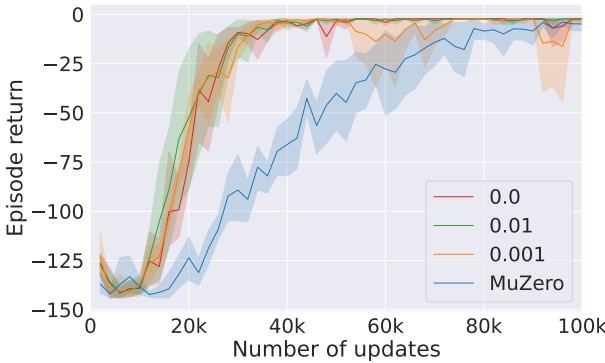

Figure 16: Ablation on the sparsity regularization coefficient $\lambda$.

Table 3: Search space reduction.

| | DoorKey | | | Sokoban | | |
|---|---|---|---|---|---|
| Easy | Normal | Hard | Easy | Normal | Hard |
| 66.7% | 75.0% | 80.0% | 46.7% | 88.9% | 95.2% |

Table 4: Training time comparison.

| Models | 4h | 8h | 12h | 16h | 20h |
|---|---|---|---|---|---|
| Ours | -116.39 | -13.00 | -3.89 | -5.30 | -2.19 |
| MuZero | -134.16 | -114.96 | -80.60 | -52.21 | -49.60 |

robustness to variations in $\lambda$. Interestingly, our method with a sparsity coefficient of $\lambda = 0.0$ seems to achieve implicit sparsity. Consequently, we employed $\lambda = 0.0$ in all primary experiments on the DoorKey and Sokoban environments.

**Search space reduction.** We describe the extent to which the MCTS search space is reduced in Table 3. We measured the action space reduction at the root node across Easy, Normal, and Hard difficulty levels within the DoorKey and Sokoban environments after 100,000 gradient steps. Table 3 illustrate the percentage reduction in search space, thereby demonstrating the efficacy of the proposed action abstraction method. Higher percentages indicate more substantial reductions.

**Training time comparison.** In addition to sample efficiency, it is crucial to assess a method under constrained computational resources. Table 4 displays the episodic returns obtained by both methods on an NVIDIA RTX 3090 after various training durations (4 hours, 8 hours, 12 hours, 16 hours, and 20 hours) in the DoorKey-Easy environment. For example, after 12 hours of training, our model achieved an episodic return of -3.89, compared to MuZero's -80.60. The wall clock time presented for training each method includes the evaluation time for completing 32 evaluation episodes every 2000 update steps, as detailed in Appendix B.1. The result show that our method is not computationally intensive relative to the baseline. Further discussion on the computational overhead of our method is provided in Appendix D.

## D ADDITIONAL DISCUSSIONS

**Challenges and approaches in identifying CSIs.** Deriving CSIs from the dynamics model has been a challenging problem. Even if we have access to the dynamics model, the context-specific independences inherent in a factored MDP remain latent and challenging to discern. An approach to approximate these CSIs could be the sample-based testing algorithm introduced in CAMPs [Chitnis et al., 2021]. However, it falls short in uncovering CSIs within a latent dynamics model and struggles with scalability in larger state and action spaces. The algorithm's time complexity further complicates its application, especially in pixel-based environments. Additionally, identifying all independence beforehand is a formidable task. In contrast, our method progressively learns CSIs as training advances.

**Learning and leveraging the action mask.** While querying the legal action might be possible if the underlying simulator is provided, it is not feasible when we consider using the learned latent dynamics model during the search procedure. Our approach involves learning and inferring an action mask for each state, applicable in the search procedure. This action mask not only delineates illegal actions but also identifies redundant action variables.

**Benefits of leveraging CSIs over policy network without action abstraction.** Consider a CMAB environment characterized by a factored action space $A = A^1 \times A^2$ with $A^1$ and $A^2$ having potential values of $\{0, 1\}$. In our CMAB environment, the consequent state is exclusively dependent on an action variable. If we consider a state $s_1$ where any action incorporating

$A^1 = 1$ transitions into a rewarding state $s_2$, whereas other actions revert to the same non-rewarding $s_1$, actions with $(1, 0)$ and $(1, 1)$ are equally optimal. Hence, the policy in standard MCTS methods like MuZero would select both actions, resulting in fewer visitation and inaccurate statistics for each action.

**Comparison to CAMPs.** CAMPs [Chitnis et al., 2021] consider a goal as a context and introduce a context-specific abstraction for each goal in preparation for planning. This approach cannot employ CSIs in each state. For instance, the method proposed in CAMPs *does not induce any action abstraction* in DoorKey environments since all action variables are required to reach the designated goal. Our work, on the other hand, is capable of inferring context-specific independence on-the-fly, while additionally leveraging action abstraction at every individual state.

**Computational overhead of training the auxiliary network.** In terms of computational efficiency, our implementation of the method demonstrates comparable performance to MuZero, with each gradient step requiring approximately the same time. While it may be differ depending on the implementation, the proposed method for learning CSI relationships end-to-end incorporates an conditional structure inference network coupled with a sparsity loss. We merely use a residual block and fully connected layers, as detailed in Appendix B.3. Notably, the architecture parallels that of both the policy and value prediction networks and the initial parts of the network are shared with the policy and value networks.