# OpenReview forum: "Efficient Monte Carlo Tree Search via On-the-Fly State-Conditioned Action Abstraction"
_auai.org/UAI/2024/Conference — UAI 2024 oral_

### Official Review · Reviewer_jmcY · 2024-02-29

**Q2-1 Originality-Novelty:** 2
**Q2-2 Correctness-Technical Quality:** 3
**Q2-5 Clarity Of Writing:** 4

**Q1 Summary And Contributions:**

This paper addresses the issue faced by agents when they have to deal with a vast set of actions, but only a subset of them is relevant at each state to determine the transition results. There were other similar approaches before, but they were all model-based. Instead, this paper builds upon Muzero, which learns the latent dynamics of the environment and does not require any prior knowledge of the environment. The authors fully analyzed their method in two environments: DoorKey and Sokoban.

**Q2-3 Extent To Which Claims Are Supported By Evidence:**

3: Good: the main claims are supported by convincing evidence (in the form of adequate experimental evaluation, proofs, (pseudo-)code, references, assumptions).

**Q2-4 Reproducibility:**

2: Fair: key resources (e.g. proofs, code, data) are unavailable but key details (e.g. proof sketches, experimental setup) are sufficiently well-described for an expert to confidently reproduce the main results.

**Q3 Main Strengths:**

1- Very detailed Experiments.
2- A good ablation study of all components in the algorithm.
3- Motivation behind the work.
4- The paper is well-written, which makes it easy to understand.

**Q4 Main Weakness:**

I will give a more complete explanation of my concerns in the comments

1- Comparison with state-of-the-art approaches.
2- As an experimental paper, we wanted to more environments.

**Q5 Detailed Comments To The Authors:**

Here are my questions and some comments about the paper:

1- I understand that the authors made an effort to thoroughly analyze the work, which appears to be the first one that does not use a known environment model. However, we would like to see how much improvement can be achieved in more challenging tasks. It seems that the authors attempted to demonstrate scalability by using larger action sets for each problem. Additionally, they introduced a more complex setting of DoorKey in the Appendix. Nevertheless, we can also apply this approach to Sokoban and incorporate an extra environment.

2- The performance comparison was conducted solely with vanilla Monte Carlo Tree Search (MCTS). My expectation was to observe a flawless performance using prior knowledge, and another method (Chitnis et al. [2021]) that leveraged a model of the environment. Although their approach may differ from yours, we are interested in seeing how it compares to the perfect performance.

3- Healthcare tasks are mentioned as one of the primary motivations behind this work. Would it be possible to see how much improvement we can get in such tasks?

4- How much does regularization impact the reconstruction loss (Eq. 7) ?

5- Function "Dec" is not defined in Eq. 7.

6- Eq. 10 seems incorrect. The policy priors should not only be the summation of probs over irrelevant actions.

7- Eq. 8 seems incorrect as well. Shouldn't it be larger than the threshold (\tau)?

8- Figure 6 is amazing. We can clearly see how action abstraction works.

9- The position of the legend should be changed in Figure 9. The figure below is hard to see.

10- Does the gap get narrower if simulation budgets keep increasing (Figure 10)?

11- Have you tried using your action abstraction with the Muzero latent dynamics model?

**Q9 Complying With Reviewing Instructions:**

Yes

---

> ### Author Rebuttal · Authors · 2024-04-05
>
> Thank you for your insightful comments. We greatly appreciate your invaluable input and address your comments as follows.
>
> ---
>
> > [Q1] The performance comparison was conducted solely with vanilla Monte Carlo Tree Search (MCTS). My expectation was to observe a flawless performance using prior knowledge, and another method (Chitnis et al. [2021]) that leveraged a model of the environment. Although their approach may differ from yours, we are interested in seeing how it compares to the perfect performance.
> >
> - Thank you for your insightful suggestions. We would like to note that Chitnis et al. (2021) utilizes conditional independence test with true environment model to find CSI relationships, which is infeasible on high-dimensional observations. Instead, we can compare to the oracle model leveraging prior knowledge on CSI relationships, which would serve as the *upper bound* of our model. We agree that this would be an interesting analysis, and we will evaluate its performance on CMAB since existing benchmarks (including DoorKey and Sokoban) do not provide ground-truth CSI relations. We again appreciate the reviewer for constructive advice, and we will incorporate this into the final manuscript.
>
> > [Q2] How much does regularization impact the reconstruction loss (Eq. 7)?
> >
> - During our experiments, we found that it does not have much impact on the quality of the reconstruction under a low value of the regularization coefficient.
>
> > [Q3] Does the gap get narrower if simulation budgets keep increasing (Fig 10)?
> >
> - We speculate that the performance gap would get narrower with an increase in simulation budgets, i.e., $B\to \infty$.
>
> > [Q4] Eq. 8 / Eq. 10
> >
> - (Eq. 8) Yes, the reviewer is correct; it is a typo. It should be $\phi_z(A)=\left\\{A^i \mid p_z^i \gt\tau\right\\} \subseteq A$ and we will fix this in the final version.
> - (Eq. 10) Sorry for the confusion. Eq. 10 is formally written as:
>
> $$
> \pi_\theta(z, \phi_z(a))=\sum_{\\{b\in \mathcal{A}\mid \phi_z(b)=\phi_z(a)\\} } \pi_\theta(z, b)=\sum_{a''\in \phi_z^c(\mathcal{A})}\pi_\theta(z, \phi_z(a), a'')
> $$
>
> - For example, if $A=[A^1, A^2, A^3]$ where the action variables are binary, $\phi_z(A)=[A^1, A^2]$, and $\phi_z(a)=(0, 0)$, then we are marginalizing over the third dimension: $\pi_\theta(z, \phi_z(a))=\pi_\theta(z, (0,0,0)) + \pi_\theta(z, (0,0,1))$. We will clarify this in the final version.
>
> > [Q5] Have you tried using your action abstraction with the Muzero latent dynamics model?
> >
> - Thank you for your intriguing suggestion. As of now, we have not explored the possibility of integrating our learned action abstraction with MuZero’s latent dynamics model. We conjecture that the performance may not be comparable with the methods due to the discrepancy between the latent state spaces of our method (which is influenced by our conditional structure inference network) and that of MuZero.
>
> > [Q6] how much improvement can be achieved in more challenging tasks / healthcare tasks
> >
> - We agree that it would be beneficial to evaluate our method in more challenging tasks or real-world scenarios (e.g., healthcare). We wish to emphasize that our approach utilizing factored actions without domain knowledge (e.g. transition structure, hierarchies of sub-actions, known environment model), was evaluated in environments with vast combinatorial action spaces. We believe our work would serve as an important stepping stone toward efficient planning utilizing the compositional structure among the variables in a more challenging real-world scenarios.
>
> > [Q7] "Dec" is not defined in Eq. 7 / The position of the legend should be changed in Fig. 9
> >
> - Sorry for the inconvenience. We will revise the manuscript to improve the clarity based on the reviewer’s comments.
>
> ---
>
> We appreciate the reviewer for the detailed feedback and valuable suggestions. We will incorporate our responses into the revised manuscript. Please let us know if you have any remaining concerns.

---

### Official Review · Reviewer_Nw3U · 2024-03-14

**Q2-1 Originality-Novelty:** 3
**Q2-2 Correctness-Technical Quality:** 3
**Q2-5 Clarity Of Writing:** 3

**Q1 Summary And Contributions:**

The performance of Monte Carlo Tree Search (MCTS) degrades under vast combinatorial action spaces. This paper considers the setting of a large factored action space. The proposed architecture reduces the number of actions to explore by learning to exploit local independencies (e.g., opening a door is irrelevant if it is already open or if you are not near).

More specifically, the authors extend the latent dynamic network of MuZero with a state-conditioned action abstraction that allows it to only consider relevant actions. The proposed auxiliary network takes the form of a mask, which is trained to hide irrelevant actions for a given embedded state (MCTS node; exploiting local independence). Importantly, the method does not require the true environment model to be known, as the abstraction is automatically learned from observations.

Experimental results demonstrate a significant benefit when the factored action space is large but contains exploitable structure.

**Q2-3 Extent To Which Claims Are Supported By Evidence:**

3: Good: the main claims are supported by convincing evidence (in the form of adequate experimental evaluation, proofs, (pseudo-)code, references, assumptions).

**Q2-4 Reproducibility:**

2: Fair: key resources (e.g. proofs, code, data) are unavailable but key details (e.g. proof sketches, experimental setup) are sufficiently well-described for an expert to confidently reproduce the main results.

**Q3 Main Strengths:**

The studied problem is very relevant, and the results positive, indicating **potential for high impact**.

The proposed idea, learning to mask irrelevant actions to improve MCTS search, makes a lot of sense conceptually. To my knowledge, this proposed approach is also **novel** (and different from, orthogonal to, state abstraction methods).

**Q4 Main Weakness:**

Experiments could be improved: more thought could be given to what is measured and what is shown in the paper. Notably, besides measuring the improved performance, I expected measuring the reduction in explored MCTS search space (cf. Q5.2, Q5.5, and to some degree Q5.1, Q5.6)

Reproducibility: while the supplemental material provides sufficient information to reproduce the proposed architecture. Releasing the relevant code (in the camera-ready version), and the specific benchmarks that were used, would be desirable.

**Q5 Detailed Comments To The Authors:**

1) In Figure 5, the y-axis. How do you obtain -150 episodic return if the Horizon is 150 and each step is -0.1? Should the worst case not be -15? Figure 10, also on DoorKey, has y-axis [0,-15].

2) Is the paragraph on "Conditional structure inference network h" not providing the same information as the paragraph on "Visualization of the state-conditioned action abstraction"?

3) In Equation 7, the regularized reconstruction loss encourages the model to only use the necessary action variables (~the M part). However, according to Appendix B.3, the sparsity regularization coeffect $\lambda = 0$ for DoorKey and Sokoban, so how does the system still learn to only use the necessary action variables? Related to this, paragraph "Training.", states "The conditional structure inference network is trained only with the reconstruction loss ...", but for DoorKey and Sokoban $\lambda$ is 0 such that the CSI network is not part of the reconstruction loss? How does it learn?

4) In DoorKey, can you pick up a key if you are not next to it? i.e., is Fig 6b exact or is picking up a key also irrelevant? The paragraph below it confused me since, in contrast, 6a predicts that opening the door is irrelevant because it is not near.

5) I was expecting an emperical study measuring how much of the MCTS search space is reduced by the proposed architecture. That seems to be missing.

6) The ablation study was not clear to me. How do you use a masked latent dynamics model without action abstraction? The mask must be learned (which is the action abstraction), no?


In appendix, computational overhead paragraph: "While it may be differ depending on" -> "may be different" or "may differ"

**Q9 Complying With Reviewing Instructions:**

Yes

---

> ### Author Rebuttal · Authors · 2024-04-05
>
> We sincerely appreciate your positive support and constructive comments. We respond to your comments below:
>
> ---
>
> > [Q1] I was expecting an empirical study measuring how much of the MCTS search space is reduced by the proposed architecture. That seems to be missing.
> >
> - We appreciate your insightful suggestions. If one action variable (key or door) is masked along the trajectory (e.g., Fig. 8), our method would reduce the search space during MCTS by approximately 80% in DoorKey-Hard. Following the reviewer’s suggestion, we will include the analysis of the search space reduction in the final version.
>
> > [Q2] In Figure 5, the y-axis. How do you obtain -150 episodic return if the Horizon is 150 and each step is -0.1? [..] Figure 10, also on DoorKey, has y-axis [0,-15].
> >
> - (Fig. 5) Sorry for the confusion. The horizon is H=1440 in DoorKey, and a worst-case episodic return is -144: there was a typo in Appendix A.1.
> - (Fig. 10) The y-axis represents the *final* performances after 100k gradient steps.
>
> > [Q3] Is the paragraph on "Conditional structure inference network h" not providing the same information as the paragraph on "Visualization of the state-conditioned action abstraction"?
> >
> - It may seem both sections discuss similar aspects of our work, they focus on different elements. “*Visualization of the state-conditioned action abstraction*” illustrates the probabilities of dependencies for all action variables across various states. On the other hand, the paragraph “*Conditional structural inference network h*” focuses on the important action variables, i.e., *pick* and *open*, and illustrate how their probability changes along the trajectory. We will integrate two paragraphs to improve clarity in the revised manuscript.
>
> > [Q4] the sparsity regularization coefficient is 0 for DoorKey and Sokoban, so how does the system still learn to only use the necessary action variables? Related to this, paragraph "Training.", states "The conditional structure inference network is trained only with the reconstruction loss ...", but for DoorKey and Sokoban \lambda is 0 such that the CSI network is not part of the reconstruction loss?
> >
> - During our experiments, we found that the results are similar when using $\lambda$ as 0 or 0.001. We observed that the initial probability outputs are low (<0.3) in the beginning of the training, and the model tends to learn higher probabilities for the necessary action variables.
> - Sorry for the confusion. If regularization coefficient is 0, then the CSI network is trained only with the *pure* reconstruction loss  (eq. 7).
>
> > [Q5] In DoorKey, can you pick up a key if you are not next to it? i.e., is Fig 6b exact or is picking up a key also irrelevant? The paragraph below it confused me since, in contrast, 6a predicts that opening the door is irrelevant because it is not near.
> >
> - In DoorKey, a single step state transition involves a sequence of inner steps where the agent first turns, then moves forward, and finally either picks up a key or opens a door if an object is present in front of it just at that moment. In Fig 6b, the agent *can* pick up the key if it does not turn, moves forward, and then performs the action of picking up the key. In contrast, in Fig 6a, the agent *cannot* open the door since it can only move into the very next cell at a time.
>
> > [Q6] (ablation study) How do you use a masked latent dynamics model without action abstraction? The mask must be learned (which is the action abstraction), no?
> >
> - In Fig. 9, “*w\o action abstraction*” denotes our method with *vanilla* MCTS for planning, i.e., we learn the mask (with latent dynamics model) but never use it for planning. The primary objective of this ablation study is to investigate the impact of action abstraction in MCTS isolated from the learning of masked dynamics.
>
> > [Q7] code
> >
> - We promise that we will make the source code publicly available after the paper is published.
>
> ---
>
> We appreciate the reviewer for the detailed feedback and valuable suggestions. We will incorporate our responses into the revised manuscript. Please let us know if you have any remaining concerns.

---

### Official Review · Reviewer_wCAN · 2024-03-18

**Q2-1 Originality-Novelty:** 3
**Q2-2 Correctness-Technical Quality:** 3
**Q2-5 Clarity Of Writing:** 3

**Q1 Summary And Contributions:**

The paper designs a method to reduce the action space in the Monte Carlo Tree Search algorithm by using an attention mechanism that completely removes specific actions based on the current state, hence reducing the branching factor (e.g., removes the 'open the door' action if we do not have a key yet).

**Q2-3 Extent To Which Claims Are Supported By Evidence:**

3: Good: the main claims are supported by convincing evidence (in the form of adequate experimental evaluation, proofs, (pseudo-)code, references, assumptions).

**Q2-4 Reproducibility:**

2: Fair: key resources (e.g. proofs, code, data) are unavailable but key details (e.g. proof sketches, experimental setup) are sufficiently well-described for an expert to confidently reproduce the main results.

**Q3 Main Strengths:**

The method is novel, technically sound, and performs very well empirically.

The experiments are very well designed, showing how the individual components of the method contribute and help to understand the method better.

**Q4 Main Weakness:**

The writing is a bit verbose, and I needed multiple passes to understand all the components. Authors refer a lot to MuZero initially, and it is difficult for someone outside the domain to follow. I recommend starting Section 3 with a high-level overview of the method before diving into the details. Also, a simple example of the entire process compared to MuZero could be helpful. The experiments later help anchor the individual components. It would be helpful if this flow had been summarized before into a simple experiment.

I encourage the authors to share their code to increase their score on reproducibility.

Authors show in Figure 10 episodic return per number of simulations. I would also expect to see episodic return per time spent. As this method has an overhead computation, how expensive it is to run should be discussed. In many cases, the budget is computation time, and we do as many MCTS runs as the time allows.

**Q5 Detailed Comments To The Authors:**

Please, see the comments above.

**Q9 Complying With Reviewing Instructions:**

Yes

---

> ### Author Rebuttal · Authors · 2024-04-05
>
> We appreciate the reviewer’s valuable efforts and instructive advice to improve the manuscript. We respond to your comments as follows:
>
> ---
>
> > [Q1]  episodic return per time spent / computation time
> >
> - We measured the wall-clock time for each gradient step for both methods in the DoorKey-Hard environment across 3 runs: It took $1.636\pm0.071$ seconds for our method and $1.607\pm0.037$ for MuZero. This implies that the evaluation in terms of the episodic return *per time spent* would not differ much from our main experiments. We agree with the reviewer that the evaluation under the computation time budget is also important, and we believe it would further strengthen our paper. We will incorporate this evaluation and discussions into the final manuscript.
>
> > [Q2] the paper structure / a high-level overview / background on MuZero
> >
> - Thanks for the valuable suggestions. We will incorporate the reviewer’s suggestions into the final manuscript, including a detailed background on MuZero, a high-level overview in the beginning of Sec. 3, and an illustrative example comparing our method and MuZero.
>
> > [Q3] code
> >
> - We promise that we will make the source code publicly available after the paper is published.
>
> ---
>
> We appreciate the reviewer for the detailed feedback and valuable suggestions. We will incorporate our responses into the revised manuscript. Please let us know if you have any remaining concerns.

---

### Official Review · Reviewer_fSiH · 2024-03-25

**Q2-1 Originality-Novelty:** 3
**Q2-2 Correctness-Technical Quality:** 3
**Q2-5 Clarity Of Writing:** 3

**Q1 Summary And Contributions:**

This paper describes the problem of learning context-specific independence within a factored action space, exploited to reduce the combinatorial branching factor of Monte Carlo tree search methods applied to MDPs for reinforcement learning (or planning).  Such a large branching factor otherwise leads to shallow trees that fail to optimize long-term discounted rewards, especially in sparse reward settings (e.g., when agents are only rewarded at the end of a game).  The solution is based on training (1) an encoder to capture latent features of high dimensional inputs (e.g., pixels from computer vision), (2) a second network to capture the transitions within that latent feature space, and (3) a third network that learns context-specific independence of action variables from the factored action space as a function of the latent feature space.  Notably, a K-step reconstruction loss is defined to train all three networks end-to-end, where a regularization term over the context-specific independence output in order to train the model to minimize the number of actions predicted to be relevant in a given context.  The trained framework is then used to predict the relevant actions under every state during MCTS so that only relevant actions are considered, minimizing the overall branching factor of the search tree.  Experimental results on two benchmarks (DoorKey and Sokoban) demonstrate the advantages of the approach over state-of-the-art approach MuZero.

**Q2-3 Extent To Which Claims Are Supported By Evidence:**

3: Good: the main claims are supported by convincing evidence (in the form of adequate experimental evaluation, proofs, (pseudo-)code, references, assumptions).

**Q2-4 Reproducibility:**

2: Fair: key resources (e.g. proofs, code, data) are unavailable but key details (e.g. proof sketches, experimental setup) are sufficiently well-described for an expert to confidently reproduce the main results.

**Q3 Main Strengths:**

S1) The combinatorial growth of factored action spaces is an important challenge for long-term planning and reinforcement learning in real-world applications, yet is understudied compared to solutions addressing the size of the agent's state space.  Advancements in this area are useful not only for single agent decision making, but could possibly be extended to multiagent settings where joint actions are inherently combinatorial of all agents.  I expect this work to be of interest to UAI attendees from the reinforcement learning and multiagent systems communities.

S2) The solution design is straightforward and well motivated.  I would encourage the authors to publish their code to enhance reproducibility, but the method could hopefully be reimplemented from the published details.

S3) I appreciated the inclusion of two different benchmarks (and three different settings within), as well as the depth of empirical analysis not focused on just reward maximization but also contributions of different parts of the solution and the quality of the context-specific independence predictions.

**Q4 Main Weakness:**

W1) Some of the results were difficult to follow.  For example, I did not understand what Figure  9 was visualizing, how SHD was measured in Figure 7a (an equation would be helpful), or how contextual bandits were used in Figure 7b.

Similarly, Table 1 comparing Reconstruction Loss seemed like an apples-and-oranges comparison since your loss includes a regularization term (that adds to the loss), whereas that is not present in MuZero.  It wasn't clear whether you reported only the rest of your loss function, or if the regularization penalty was included.

The mathematical notation was also difficult to follow at times (examples provided below).

W2) I was a little surprised to see the probabilities of dependencies for actions to always be so binary in Figures 6 and 8.  I would have expected some values to be a little above 0 or a little below 1.  Does this maybe indicate that the problem was rather simple to learn the action dependencies, and what would happen in more complicated settings where it is not so obvious when actions are relevant (e.g., the presence of a key or door are easy to capture visually).  Was this a function of your very low threshold (\tau = 0.01), and what would happen if you increased this threshold?

W3) It would be helpful if the authors included a link to their code in the camera-ready version (if accepted) for the sake of reproducibility.

**Q5 Detailed Comments To The Authors:**

In Equation 8, \phi_z(A) appears to be a set of action variables, reduced by \phi_z from the full set A.  Then in Equation 9, is phi_z(a) a single action (since a lower case a is used)?   The notation implies that \phi_z is a function that transforms an input, but I am assuming that is not the case here (since \phi is an operator on a set not a single action variable)?  I would suggest using something like a' instead of \phi_z(a), then replace the part under the argmax with $a' in  \phi_z(A)$.

I also didn't understand the justification for Equation 10 -- I assume you are summing over all action variables not in \phi_z(A) [since \phi^c_z is used], but I do not understand why you would marginizalize over \phi^c_z.

**Q9 Complying With Reviewing Instructions:**

Yes

---

> ### Author Rebuttal · Authors · 2024-04-05
>
> We sincerely appreciate your efforts and constructive comments to improve the manuscript. We respond to your comments below.
>
> ---
>
> > [Q1] Fig 9 / Fig 7 (SHD, contextual bandit) / Table 1
> >
> - (Fig 7) Structural hamming distance (SHD) is a commonly used metric to measure the difference between two directed acyclic graphs $\mathcal{G}_1$ and $\mathcal{G}_2$, defined as $\\|A_1-A_2\\|_F^2$ where $A_1$ and $A_2$ are the adjacency matrices of  $\mathcal{G}_1$ and $\mathcal{G}_2$, respectively. We designed the contextual multi-armed bandit (CMAB) scenario to investigate the relevance between the task performance and CSI discovery. The detailed parametrization of CMAB is provided in Appendix A.3.
> - (Fig 9) Recall that the model is trained with multiple losses, e.g., policy, value and reconstruction losses, “*w\o recon*” denotes the method trained without reconstruction loss. “*w\o action abstraction*” denotes our method with vanilla MCTS for planning, i.e., a conditional structure inference network is trained with latent dynamics model but never used for planning. Overall, the ablation study shown in Fig 9 implies (a) the reconstruction loss is crucial for the model training and (b) the effectiveness of state-conditioned action abstraction for MCTS.
> - (Table 1) Sorry for the confusion. Reported numbers for our method in Table 1 is the *pure* reconstruction loss which does not include the regularization loss term. We will clarify this in the final version.
>
> > [Q2] Notations: $\phi_z(A)$ and $\phi_z(a)$ (Eq. 8-9)
> >
> - Sorry for the confusion. Recall that $A=[A^1, \cdots, A^n]$ denotes the set of action *variables*, where $a=(a^1, \cdots, a^n)\in\mathcal{A}^1\times \cdots \times \mathcal{A}^n=\mathcal{A}$ denotes their *values*. Similarly, $\phi_z(A)$ and $\phi_z(a)$ denote the abstract action variables and their values. For example, if $\phi_z(A) = [A^1, A^2]$, then $\phi_z(a) = (a^1, a^2)\in \mathcal{A}^1\times \mathcal{A}^2=\phi_z(\mathcal{A})$. Here, the argmax over $\phi_z(a)$ can be equivalently written as the argmax over $a'\in \phi_z(\mathcal{A})$. Thanks for the suggestion!
>
> > [Q3] Eq. 10
> >
> - Sorry for the confusion. The summation over $\phi^c_z(a)$ in Eq. 10 is formally written as:
>
> $$
> \pi_\theta(z, \phi_z(a))=\sum_{\\{b\in \mathcal{A}\mid \phi_z(b)=\phi_z(a)\\} } \pi_\theta(z, b)=\sum_{a''\in \phi_z^c(\mathcal{A})}\pi_\theta(z, \phi_z(a), a'')
> $$
>
> - For example, if $A=[A^1, A^2, A^3]$ where the action variables are binary, $\phi_z(A)=[A^1, A^2]$, and $\phi_z(a)=(0, 0)$, then we are marginalizing over the third dimension: $\pi_\theta(z, \phi_z(a))=\pi_\theta(z, (0,0,0)) + \pi_\theta(z, (0,0,1))$. We will clarify this in the final version.
>
> > [Q4] the probabilities of dependencies for actions in Fig 6, 8 [..] what would happen in more complicated settings where it is not so obvious when actions are relevant (e.g., the presence of a key or door are easy to capture visually). Was this a function of your very low threshold (\tau = 0.01), and what would happen if you increased this threshold?
> >
> - We found that the probabilities of dependencies tend to converge to binary as the training proceeds, and the qualitative examples in Fig 6, 8 are the output of our model *after* the training converged and achieved near-optimal performance. In more complex settings where it is hard to determine the relevant actions, e.g., noisy observations, we speculate that the probabilities of dependencies may not converge to binary.
> - For training the latent dynamics model (Fig. 2a), we use $\tau=0.5$. For the action abstraction during MCTS (Fig. 2b), we use a conservative threshold $\tau=0.01$ since the prediction of the dependencies could be inaccurate, e.g., in the early stage of the training. Thus, using a low threshold for MCTS does not impact the training of the conditional structure inference network.
>
> > [Q5] code
> >
> - We promise that we will make the source code publicly available after the paper is published.
>
> ---
>
> We appreciate the reviewer for the detailed feedback and valuable suggestions. We will incorporate our responses into the revised manuscript. Please let us know if you have any remaining concerns.

---

### Meta-Review · Area_Chair_fEbC · 2024-04-21

The paper describes a new technique Monte Carlo Tree Search technique that abstracts out action features conditioned on the current state.  This can reduce significantly the action space in factored action space.  The reviewers unanimously recommend acceptance of the paper.  This represents an important advance for scaling Monte Carlo Tree Search in problems with factored actions.